# Selective hydrogenation of nitro compounds to amines by coupled redox reactions over a heterogeneous biocatalyst

Daria Sokolova [1], Tara C. Lurshay [1,2], Jack S. Rowbotham [1,3], Georgia Stonadge[1], Holly A. Reeve [1,2], Sarah E. Cleary [1,2] ✉, Tim Sudmeier [1] ✉ & Kylie A. Vincent [1] ✉

Cleaner synthesis of amines remains a key challenge in organic chemistry because of their prevalence in pharmaceuticals, agrochemicals and synthetic building blocks. Here, we report a different paradigm for chemoselective hydrogenation of nitro compounds to amines, under mild, aqueous conditions. The hydrogenase enzyme releases electrons from $H_2$ to a carbon black support which facilitates nitro-group reduction. For 30 nitroarenes we demonstrate full conversion (isolated yields 78 – 96%), with products including pharmaceuticals benzocaine, procainamide and mesalazine, and 4-aminophenol – precursor to paracetamol (acetaminophen). We also showcase gram-scale synthesis of procainamide with 90% isolated yield. We demonstrate potential for extension to aliphatic substrates. The catalyst is highly selective for reduction of the nitro group over other unsaturated bonds, tolerant to a wide range of functional groups, and exhibits excellent stability in reactions lasting up to 72 hours and full reusability over 5 cycles with a total turnover number over 1 million, indicating scope for direct translation to fine chemical manufacturing.

Efficient and sustainable routes to synthesis of amines remain in high demand for the production of pharmaceuticals (Fig. 1A) and agrochemicals as well as other areas of chemical manufacturing. This has led to a wide range of developments in selective methods for amine synthesis[1–3], including various biocatalytic approaches[4–8]. The reduction of nitro-groups is a common synthetic route to amines, and is a key target for greener synthetic protocols because the available routes are dominated by use of stoichiometric reductants or precious-metal hydrogenations which often lack functional group selectivity (Fig. 1B)[9–12]. Recent developments have focused on organocatalysts, or the more abundant first row transition metals as either heterogeneous or homogeneous hydrogenation catalysts (including for transfer hydrogenations using formic acid, hydrazine or $NaBH_4$), and have led

to some improvements in functional group tolerance[9,13,14]. Although biocatalytic reduction of unsaturated bonds is often viewed as an environmentally friendly and more selective alternative to metal-catalysed hydrogenations in the pharmaceutical sector, biocatalytic strategies are still in the early stages of development for the 6-electron reduction of nitro groups. Nitro reductions by flavin-containing nitroreductase enzymes typically rely upon multiple equivalents of glucose to recycle the costly redox cofactor, NAD(P)H, and with few exceptions[15], often fail to progress beyond the N-hydroxylamine intermediate, although this can be mitigated by photocatalysis[16] or addition of a co-catalyst such as $V_2O_5$[17–19].

A new concept is emerging for an 'electrochemical hydrogenation' mechanism in heterogeneous catalysis, whereby $H_2$ oxidation at

[1]Department of Chemistry, University of Oxford, Inorganic Chemistry Laboratory, South Parks Road, Oxford OX1 3QR, UK. [2]Present address: HydRegen Limited, Centre for Innovation and Enterprise, Begbroke Science Park, Oxford OX5 1PF, UK. [3]Present address: Department of Chemistry, University of Manchester, Manchester Institute of Biotechnology, Manchester M1 7DN, UK. ✉e-mail: sarah@hydregenoxford.com; tim.sudmeier@gmail.com; kylie.vincent@chem.ox.ac.uk

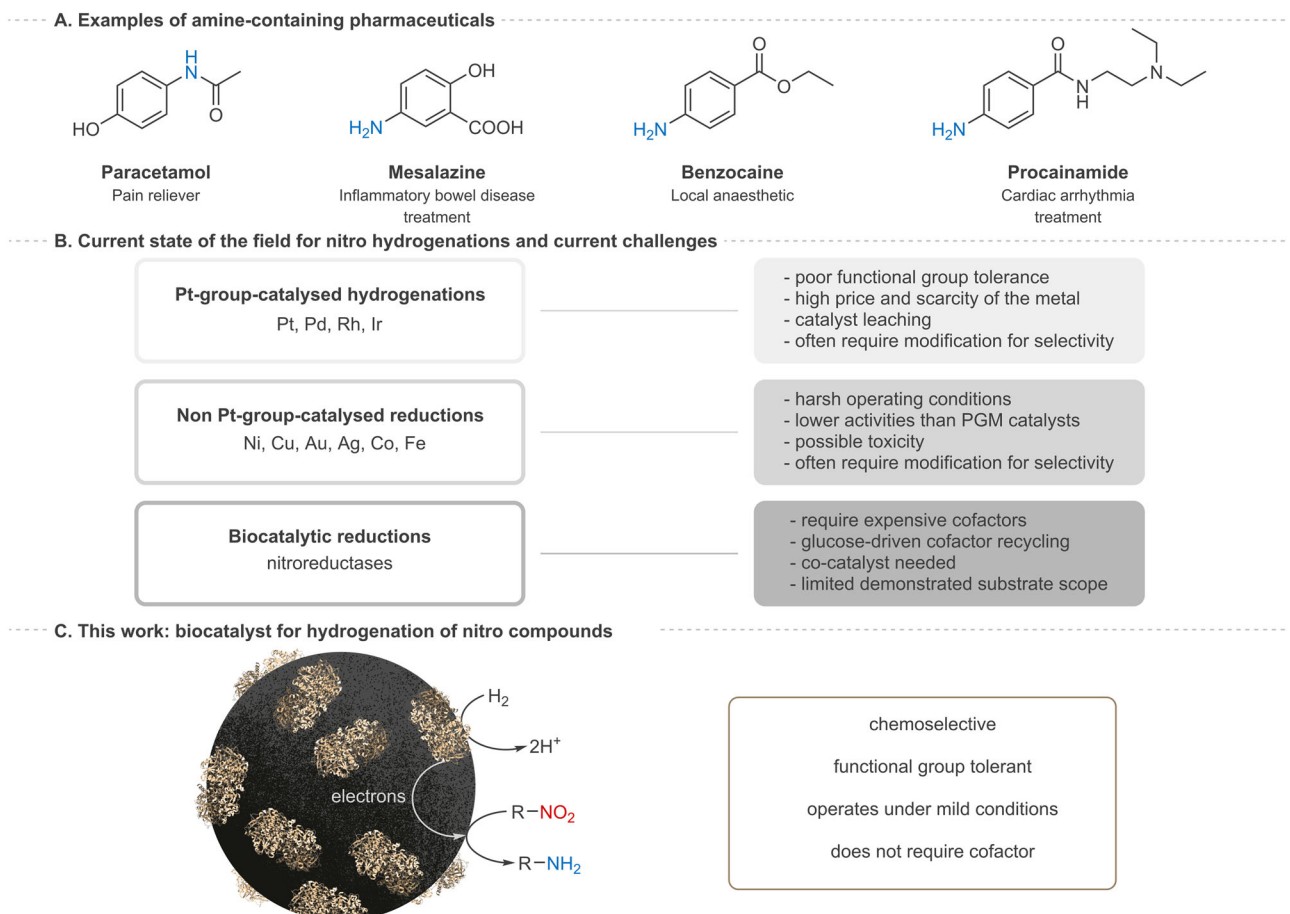

**Fig. 1 | Current state of the art and catalytic approach exploited in this work.**
**A** Examples of amine pharmaceuticals relevant to this work. **B** Currently-used
methods for the reduction of nitro compounds to generate amines. **C** Hydrogenase
enzyme (gold) immobilised on the surface of carbon black particle as a catalyst for
efficient chemoselective synthesis of amines (this work).

---

one metal site is coupled to a reduction process at a separate, but
electronically linked, catalytic site[20]. This was recently demonstrated
as the dominant mechanism in nitrite hydrogenation over a PdCu alloy
on carbon[20]. It is known that organonitro groups can be electro-
chemically reduced to the amine at a carbon electrode surface in
aqueous electrolyte, suggesting the possibility of harnessing an 'elec-
trochemical hydrogenation' mechanism in reduction of nitro
compounds[21]. In fact, it has been noted recently that nitro group
hydrogenations at palladium on carbon (Pd/C) may actually proceed
via an electrochemical mechanism whereby $H_2$ oxidation occurs at
active sites on the Pd and provides electrons for reduction of the nitro
compound at the carbon support[22]. However the presence of Pd
nanoparticles under $H_2$ can lead to a range of unwanted side reactions
on a target substrate, including hydrogenation of other unsaturated
bonds or dehalogenation, while sulfur-containing substituents may
poison the precious metal catalyst[9,10]. In contrast, hydrogenase
enzymes are able to split $H_2$ at a buried active site which is inaccessible
to larger organic substrates, and release electrons, via a chain of iron-
sulfur clusters, to the protein surface[23]. In this work we show that
hydrogenase on carbon can be exploited as a *selective* catalyst for
activation of $H_2$, to give Pd-free 'electrochemical hydrogenation' of
nitro compounds (Fig. 1C).

## Results and discussion

### Development of the catalytic system

We have shown previously that the nickel-iron hydrogenase enzyme
adsorbs readily onto a carbon black support and is able to channel

electrons from $H_2$ oxidation into the carbon where they can be taken
up by a co-adsorbed reductase enzyme[24–27]. For this study we select the
robust and $O_2$-tolerant nickel-iron (NiFe) hydrogenase 1 from *Escher-
ichia* (*E.*) *coli* (Hyd-1), which is purified readily following modest over-
expression in the native host organism[28].

The onset potential for reduction of nitrobenzene (**1**) (Fig. 2A,
Supplementary Fig. 48), a model aromatic nitro compound, on a gra-
phite electrode in aqueous medium at pH 6.0 commences at −0.113 V
vs the standard hydrogen electrode (SHE; all subsequent potentials are
quoted vs this reference). At pH 6.0, 1 bar $H_2$, the potential of the
proton/dihydrogen couple, $E'(2H^+/H_2)$ is −0.355 V. The fact that the
onset of nitrobenzene reduction is positive of $E'(2H^+/H_2)$ means that
reduction of the nitroaromatic compound by $H_2$ is thermodynamically
feasible, i.e. the overall reaction has a negative free energy change.
Hyd-1 has a small over-potential relative to $E'(2H^+/H_2)$[29], with the onset
potential for $H_2$ oxidation lying at −0.296 V (Fig. 2B). We therefore
hypothesised that a catalyst comprising Hyd-1 immobilised on carbon
black particles (Hyd-1/C) should be able to carry out the hydrogenation
of nitrobenzene, where the reduction of the nitro group would occur
at the carbon surface, akin to an electrochemical half reaction, using
electrons supplied from $H_2$ oxidation by the hydrogenase as shown
in Fig. 1C.

We therefore tested the feasibility of nitrobenzene hydrogenation
using a Hyd-1/C catalyst. Hyd-1 was immobilised by direct adsorption
onto carbon black which we have previously shown is suitable for
direct electron-exchange with Hyd-1[30,31]. After 12 hours of reaction
under $H_2$ flow, we observed full conversion of 10 mM nitrobenzene to

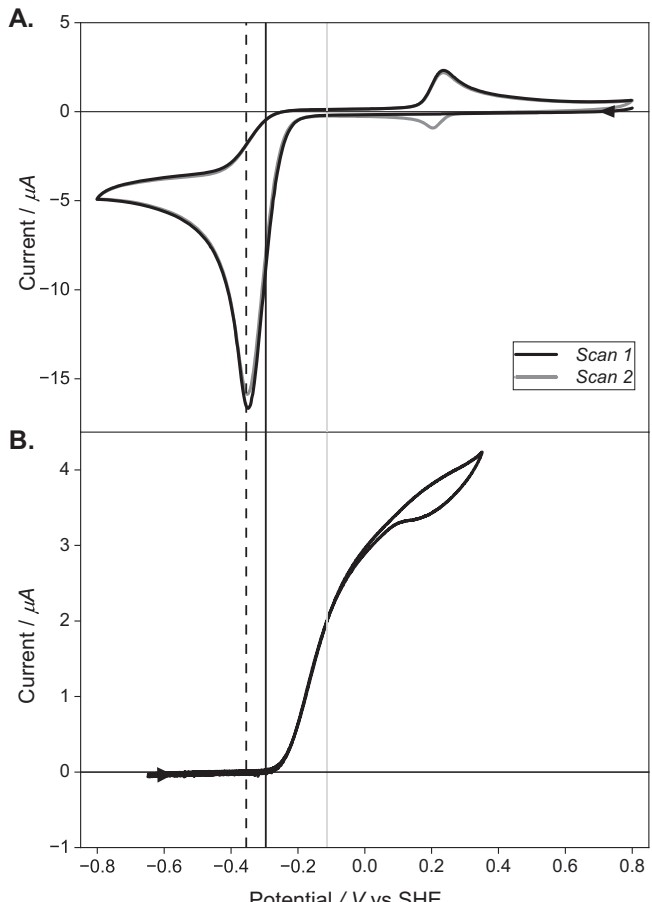

**Fig. 2 | Onset potential for nitrobenzene reduction and H₂ oxidation on a carbon electrode at 25 °C, pH 6.0.** Cyclic voltammograms for **A**: nitrobenzene at a stationary graphite electrode under a N₂ atmosphere, scan rate 10 mV/s; and **B**: a film of Hyd-1 adsorbed onto the electrode under a H₂ atmosphere with electrode rotation at 3000 rpm, scan rate 1 mV/s. Potentials are quoted vs the standard hydrogen electrode, SHE. Dashed vertical line: potential of the 2H⁺/H₂ couple at the experimental conditions, $E'(2H^+/H_2)$; solid black vertical line: measured onset potential for H₂ oxidation by Hyd-1; solid grey vertical line: measured onset for nitrobenzene reduction (see Supplementary Table 6).

aniline (**1a**) with no side products. Control experiments confirm that neither Hyd-1 nor carbon particles alone show this reactivity (Supplementary Fig. 3). These results encouraged us to explore a wide range of aromatic nitro compounds to understand the substrate scope, functional group tolerance, and chemoselectivity of the Hyd-1/C catalyst, as summarised in Fig. 3. All nitrobenzene derivatives shown in Fig. 3 were fully hydrogenated to the corresponding amine by Hyd-1/C at 1 bar H₂. Some substrates required 10% v/v of MeCN as a co-solvent to overcome the solubility issues, and for some the length of reaction time or catalyst loading were increased to facilitate full conversion (Section IV of the Supplementary Information). These results demonstrate the high tolerance of this biocatalyst system to different substituents on the aromatic ring.

### Exploration of the substrate scope

Halogenated substrates **8–10**, and **15–16** were selected to test the ability of Hyd-1/C to hydrogenate the nitro-group without dehalogenation. Promisingly, complete, selective conversion to the amine products with no loss of the halogen substituent (Cl, Br, I) was observed in all cases (Supplementary Figs. 13–15, 20–21). Additionally, no side reduction was observed for substrate **17**, which is often the case for the reduction of benzylic alcohols and their derivatives using

Pd/C (Supplementary Fig. 22)[32]. Thiolate moieties are known to poison precious metal-based catalysts, such as Pd/C[33], but full conversion of substrate **25** was achieved in 24 hours using the Hyd-1/C catalyst (Supplementary Fig. 30).

Selectivity of Hyd-1/C for hydrogenation of the nitro group was demonstrated with substrates **18, 21, 26-28** for which full conversion of the nitro group to the amine was observed, with no evidence for reduction of ketone, aldehyde, alkene, alkyne or nitrile groups (Supplementary Figs. 23, 26, 31–33). Substrate **28** required higher catalyst loading and pH 8.0 to suppress the side reaction of alkyne hydration. The selectivity is consistent with the clean linear sweep voltammograms observed for electrochemical reduction of these nitroarenes at carbon.

Sterically hindered substrates **2, 5, 8, 11** and **29** and substrates with the bulky *tert*-butyl group in the *ortho-* or *para-* position (**23** and **24**), were fully converted to the corresponding aniline derivatives, although some required higher catalyst loadings and/or extended reaction time (Fig. 3 and Supplementary Figs. 7, 10, 13, 16, 28, 29, 34).

Substrates with two nitro groups (**11–13**) were found to be reduced completely to the corresponding di-amine using the Hyd-1/C catalyst under the experimental conditions employed (Supplementary Figs. 16–18).

We next demonstrated the use of the catalyst on nitro compounds which are precursors to pharmaceuticals. Reduction of substrate **7** gives 4-aminophenol, which requires only a simple acetylation of the amine to form the widely-used pain reliever, paracetamol (acetaminophen). Hydrogenation of substrate **20** produces the local anaesthetic, benzocaine. Reduction of substrate **30** generates the important drug molecule, mesalazine, which is used to treat inflammatory bowel disease and features in the World Health Organisation List of Essential Medicines[34].

### Isolation of products and upscale synthesis

Having demonstrated Hyd-1/C as a versatile catalyst for nitroarene reductions, we now focus on reaction-scale-up and isolation of products, summarised in Fig. 4. For most of the substrates, the corresponding amines were isolated by simple extraction with organic solvent without any further purification (see Section V of the Supplementary Information) with 78–96% yield (Fig. 4A). For the highest yielding product, **29a**, this represents $2.22 \times 10^5$ turnovers of Hyd-1 during the 24-hour reaction. Some loss of product is likely to occur during the workup due to the relatively small scale of these reactions. For chemically unstable products (**18a** and **25a**) the yields were determined by ¹H-NMR spectroscopy with the use of an internal standard.

To demonstrate scalability of the biocatalytic system for a pharmaceutically-relevant product, we chose reduction of *N*-(2-(diethylamino)ethyl)-4-nitrobenzamide (**31**) to procainamide (**31a**) – medication for treatment of cardiac arrhythmia[35]. The precursor **31** was synthesised in one step from the commercially available 4-nitrobenzoyl chloride and *N,N*-diethylethylenediamine (Fig. 4C) and then was subsequently hydrogenated using Hyd-1/C to yield 1.10 g of **31a** with 96% purity (90% yield). This clearly indicates scalability, and potential applicability of this system in the production of fine chemicals and their precursors.

### Exploration of the mechanism and catalyst recycling

To understand aspects of the mechanism of nitro hydrogenation we undertook further experiments with **1** as a model substrate. Figure 4B presents ¹H-NMR traces of the reaction progress of nitrobenzene hydrogenation, carried out at ambient H₂ pressure, in the presence of the Hyd-1/C catalyst. After 30 minutes, signals corresponding to the *N*-phenylhydroxylamine (**1b**) intermediate were observed. By 1 hour, the starting nitro compound **1** was fully consumed, giving **1b** with traces of **1a**. By 12 hours, complete conversion to the aniline was achieved.

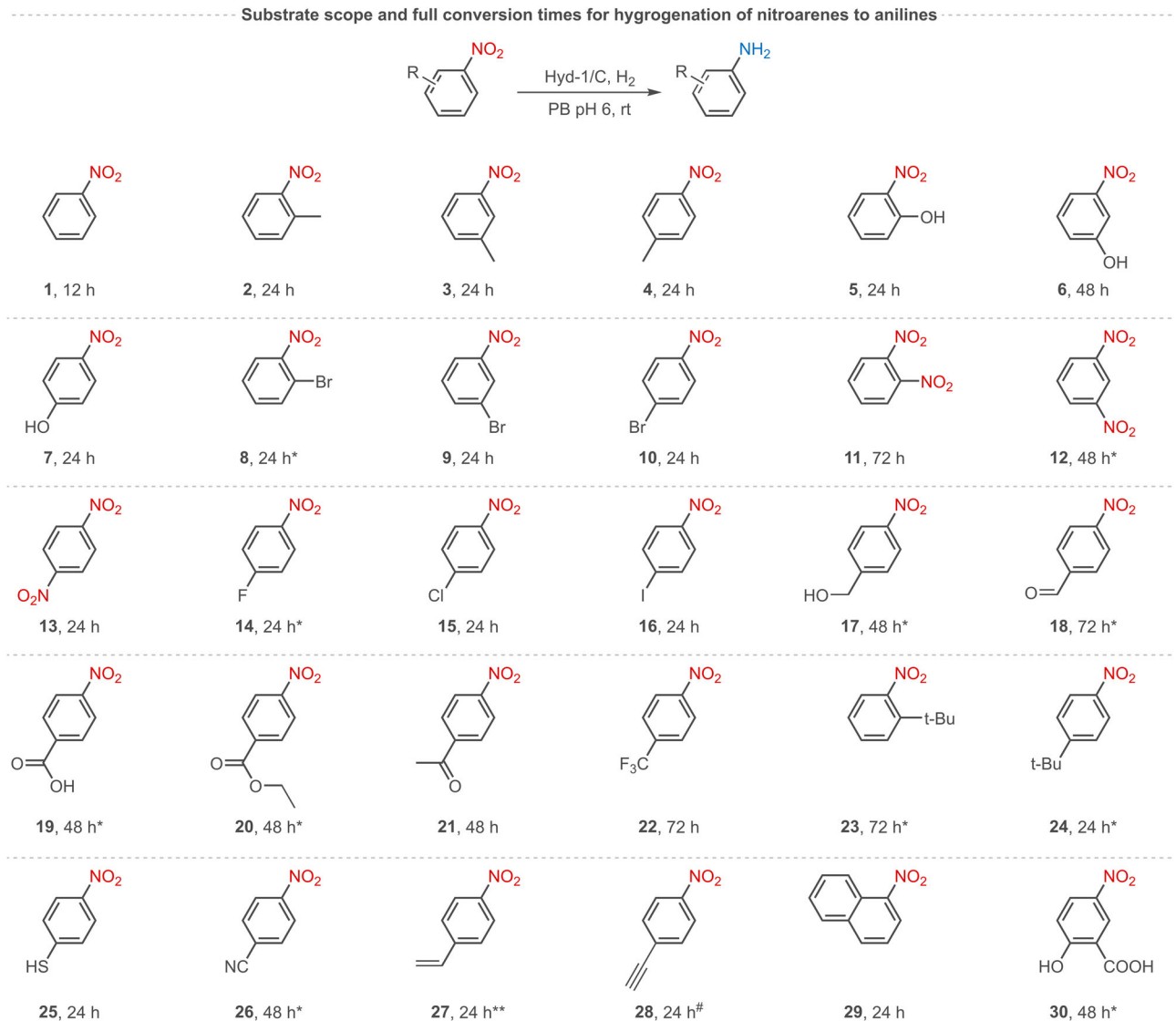

Fig. 3 | Substrate scope of hydrogenation reactions achieved with the Hyd-1/C catalytic system (unless specified, single catalyst loading: 1.06 mg of C, 26 μg of Hyd-1 per reaction) at 10 mM concentration of substrate, 2 mL reaction volume, 0% or 10% v/v of MeCN in sodium phosphate buffer (PB, 50 mM, pH 6.0, unless stated otherwise), room temperature, 1 bar H₂. *Double catalyst loading. **Quadruple catalyst loading. #Double catalyst loading, pH 8.0.

These results indicate that the reduction of nitrobenzene using Hyd-1/C proceeds via four-electron reduction to the *N*-phenylhydroxylamine intermediate. This is also corroborated by closer examination of the electrochemistry of nitrobenzene in aqueous solvent. Cyclic voltammograms of **1** (Fig. 2A) reveal that after the first cathodic sweep, a new, reversible redox couple, centred at *ca* + 0.21 V, appears on the return and subsequent sweeps. By comparison with analytical standards (Supplementary Figs. 44 and 45), this new wave is assigned to the two-electron oxidation of **1b** to nitrosobenzene, which is subsequently re-reduced on the reverse sweep, providing further evidence that **1b** is indeed produced at the potential of the nitrobenzene reduction wave. Together these results support a mechanism for the Hyd-1/C catalyst whereby nitrobenzene is first reduced by four electrons to the *N*-phenylhydroxylamine derivative, which can be subsequently reduced to the aniline. This is further evidenced in a catalyst recycling experiment, made possible because of the heterogeneous nature of our biocatalytic system (Supplementary Fig. 43). Full conversion of nitrobenzene to aniline was recorded over five cycles of recovery/re-use, representing a total turnover number (TTN) for Hyd-1 of 1.16 × 10⁶. In subsequent re-use cycles, the percentage of *N*-phenylhydroxylamine

intermediate increased gradually, although the starting material **1** was still fully consumed over 13 cycles in total. The small scale of these reactions (2 mL reaction volume) means that catalyst loss during each recovery cycle is more significant than it would be in larger scale reactions.

Supplementary Fig. 48 shows the onset potentials for the reduction of a selection of nitroarenes under aqueous conditions. All have an onset potential for reduction which is positive of *E*′(2H⁺/H₂), and also positive of the onset for H₂ oxidation by Hyd-1. We also tested the aliphatic nitro compound, 1-nitrohexane (**32**), which has a more negative onset potential of −0.313 V. Attempts to reduce **32** using the Hyd-1/C catalyst gave no product (Supplementary Fig. 77). These data allow us to set a tentative lower limit for the potential of nitro substrates which could be hydrogenated by Hyd-1/C without application of more forcing conditions.

## Expansion of the catalytic approach to aliphatic nitro compounds
We hypothesised that replacing Hyd-1 with another NiFe hydrogenase from *E. coli*, Hydrogenase 2 (Hyd-2), which shows no overpotential for

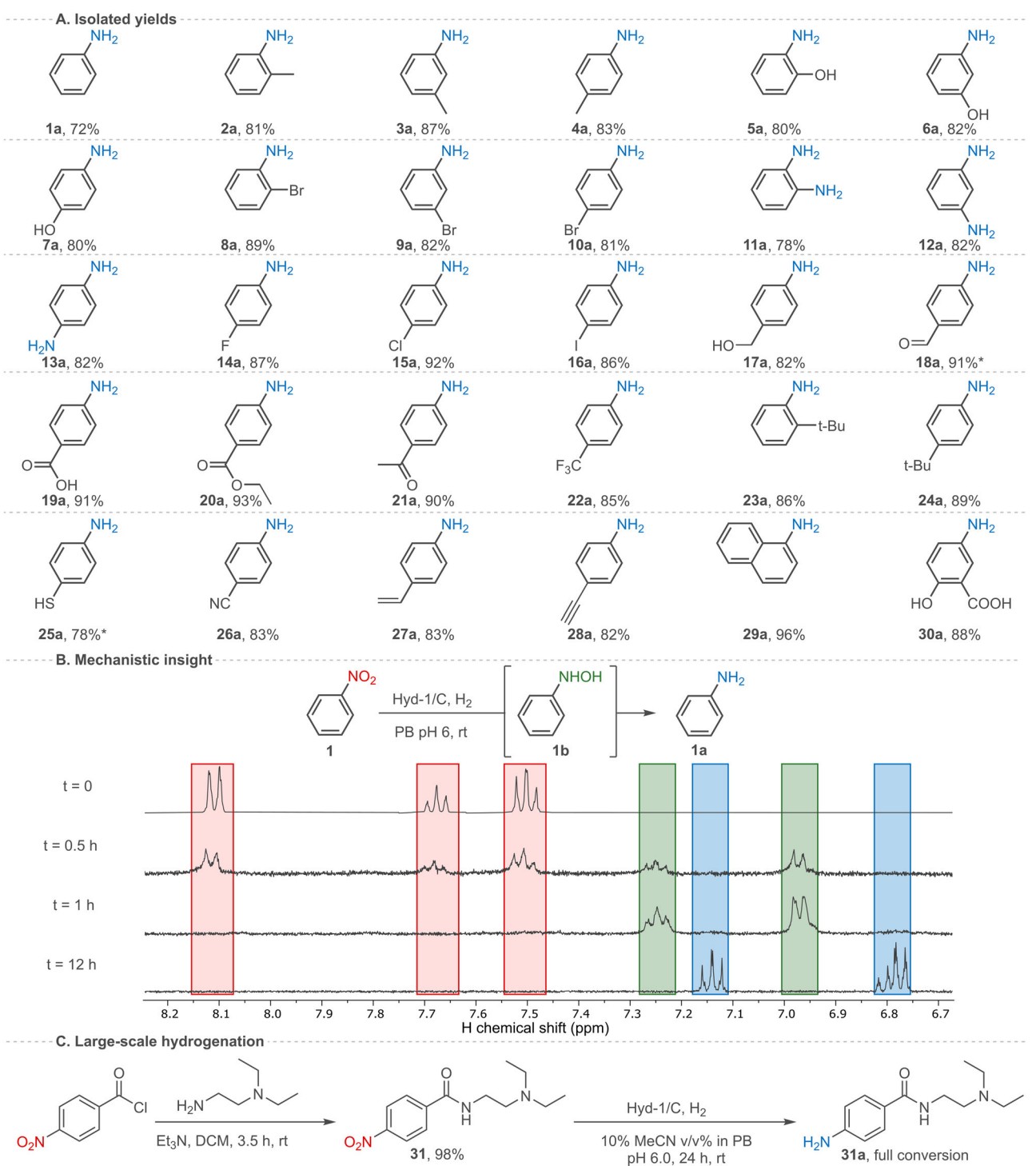

**Fig. 4 | Isolated yields, mechanistic insight and large-scale synthesis of procainamide. A** Isolated yields (%) for **1a–30a**. Conditions: as in Fig. 3 caption. *[1]H-NMR yields. **B** [1]H-NMR traces of hydrogenation of **1** at indicated time points. Conditions: Hyd-1/C (1.06 mg of C, 26 μg of Hyd-1), 10 mM **1**, 2 mL reaction volume, PB (50 mM, pH 6.0), room temperature, 1 bar $H_2$. Traces for **1**, **1b**, and **1a** are labelled with red, green, and blue, respectively. **C** Synthesis of substrate **31** and its hydrogenation on a gram-scale. Conditions: Hyd-1/C (264 mg of C, 6.63 mg of Hyd-1), 10 mM **31**, 500 mL reaction volume, 10% MeCN v/v in PB (50 mM, pH 6.0), room temperature, 1 bar $H_2$.

$H_2$ oxidation, i.e. operating at $E'(2H^+/H_2)$[29], might allow us to extend this catalytic approach to more challenging substrates. We were therefore encouraged to test whether Hyd-2/C was able to hydrogenate the aliphatic nitro compound **32**. In a 48-hour reaction, **32** was fully converted to the hexylamine (**32a**) over the Hyd-2/C catalyst (Supplementary Fig. 78), demonstrating that there is scope for expanding the 'hydrogenase on carbon' catalyst concept to aliphatic nitro compounds.

Inspired by the concept of electrochemical hydrogenation in heterogeneous catalysis, we have established a hybrid bio-chemo catalyst, which operates entirely via an electrochemical (coupled redox) mechanism. We show that this gives an easy-to-use, highly versatile

catalyst for the synthesis of amines via hydrogenation of aromatic nitro compounds under mild conditions. The catalyst comprises a carbon black supported NiFe hydrogenase (Hyd-1) which enables use of $H_2$ at atmospheric pressure as an atom-efficient reductant, without need for a co-catalyst or cofactor. We confirmed tolerance of the biocatalytic system to a wide range of functional groups by hydrogenating various derivatives of nitrobenzene to the corresponding aniline, with isolated yields from 78% to 96%. The catalyst is fully recyclable over 5 reaction cycles, and is applicable for a gram-scale hydrogenation, as demonstrated with procainamide synthesis. We further show that this catalytic approach can be extended to hydrogenation of aliphatic nitro compounds by utilising another NiFe hydrogenase (Hyd-2), which we confirm by hydrogenation of 1-nitrohexane as a model substrate. Overall, this work represents a valuable addition to the suite of approaches available for amine synthesis.

## Methods
### Chemicals
Buffer salts and HPLC grade solvents were purchased from SIGMA-ALDRICH. Deuterated solvents were purchased from THERMO SCIENTIFIC (CDCl$_3$, 99.8% D; MeOH-d$_4$, ≥99.8% D), SIGMA-ALDRICH (D$_2$O, 99.9% D), and VWR CHEMICALS (DMSO-d$_6$, 99.8% D).

Nitrobenzene (99%) was purchased from ALFA AESAR; N-phenylhydroxylamine (≥95%), nitrosobenzene (≥97%), 4-nitrotoluene (99%), 4-nitrophenol (≥99%), 1,2-dinitrobenzene (97%), 1,3-dinitrobenzene (97%), 1,4-dinitrobenzene (98%), 1-fluoro–4-nitrobenzene (99%), 1-chloro-4-nitrobenzene (99%), 1-nitronaphtalene (99%), benzyl viologen dichloride (97%), N,N-diethylethylenediamine (≥99%), triethylamine (≥99%), 3-chloroperbenzoic acid (≤77%), 1-nitrohexane (98%), hexylamine (99%), and 4-nitrobenzoyl chloride (98%) were purchased from SIGMA-ALDRICH; 2-nitrotoluene (99%), 3-nitrotoluene (95%), 2-nitrophenol (95%), 3-nitrophenol (99%), 1-bromo-2-nitrobenzene (98%), 1-bromo-3-nitrobenzene (99%), 1-bromo-4-nitrobenzene (95%), 1-iodo-4-nitrobenzene (95%), 4-nitrobenzyl alcohol (99%), 4-nitrobenzaldehyde (98%), 4-nitrobenzoic acid (98%), ethyl 4-nitrobenzoate (95%), 4′-nitroacetophenone (95%), 4-nitrobenzotrifluoride (98%), 1-tert-butyl-2-nitrobenzene (95%), 1-tert-butyl-nitrobenzene (95%), 4-nitrothiophenol (90%), 4-nitrobenzonitrile (97%), 1-ethynyl-4-nitrobenzene (95%), 5-nitrosalicylic acid (95%) were purchased from FLUOROCHEM; 4-nitrostyrene (98%) was purchased from THERMO SCIENTIFIC. Product standards 4-aminostyrene (96%) and 4-aminothiophenol (97%) were purchased from FLUOROCHEM. Carbon black (carbon, C) BLACK PEARLS® 2000 was purchased from CABOT.

All the chemicals and solvents were used as received without prior purification, except for 4-aminothiophenol, which was purified by column chromatography (silica gel, hexane/EtOAc = 5:1, $R_f$ = 0.31). All aqueous solutions were prepared with deoxygenated MilliQ water (Millipore, 18 MΩcm).

### Methods of analysis
$^1$H-NMR spectra were recorded at 400 MHz on a BRUKER AVANCE III HD NANOBAY NMR spectrometer equipped with a 9.4 T magnet. Chemical shifts of $^1$H-NMR spectra (measured at 298 K) are given in ppm by using residual solvent signals as references (CDCl$_3$: 7.26 ppm; D$_2$O: 4.79 ppm; DMSO-d$_6$: 2.50 ppm; methanol-d$_4$: 4.78 ppm and 3.31 ppm). Coupling constants (J) are reported in Hertz (Hz). Standard abbreviations indicating multiplicity were used as follows: s (singlet), brs (broad singlet), d (doublet), dd (doublet of doublets), t (triplet), q (quadruplet), m (multiplet). $^1$H-NMR signals for samples in 10% D$_2$O in sodium phosphate buffer (PB, concentration and pH are specified in each case) were measured with the water suppression method. NMR-based yields reported by analysis of $^1$H-NMR signals measured in 10% D$_2$O in PB with a quantitative water suppression method in the presence of internal standard (4-nitrophenol).

Analytical thin-layer chromatography (TLC) was performed on MERCK silica gel 60 F254 aluminium plates, which were analysed by fluorescence detection with UV-light (λ = 254 nm, [UV]) or after exposure to standard staining reagents and subsequent heat treatment. The following staining solution was used: potassium permanganate [KMnO$_4$] (1.5 g potassium permanganate, 10 g potassium carbonate, 1.25 mL sodium hydroxide (10% aqueous solution) in 200 mL water). Purification by column chromatography was performed using Geduran® Si 60 (40–63 μm) purchased from SIGMA-ALDRICH.

### Production and purification of enzymes
The Hydrogenase-1 (Hyd-1) was produced according to the reported protocol and purified using a Ni-affinity column[28]. The yield of Hyd-1 was ~0.5 mg of protein per liter of culture. The activity of the overexpressed Hyd-1 was verified using a spectrophotometric assay measuring the hydrogenase-mediated reduction of benzyl viologen dichloride in an H$_2$-saturated solution. The specific activity of $(32 \pm 1)$ nmol·min$^{-1}$·mg$^{-1}$ was measured on three independent hydrogenase-mediated benzyl viologen dichloride reduction assays. Another batch of Hyd-1 (1.9 mg of protein per liter of culture, similar activity to the previous batch) was prepared according to the same procedure to run a large-scale experiment.

The Hydrogenase-2 (Hyd-2) enzyme was produced according to the reported protocol and purified using a Ni-affinity column[36]. The yield of Hyd-2 was ~0.2 mg of protein per liter of culture. The activity of the overexpressed Hyd-2 was verified using a spectrophotometric assay measuring the hydrogenase-mediated reduction of benzyl viologen dichloride in an H$_2$-saturated solution. The specific activity of $(4.18 \pm 0.2)$ μmol·min$^{-1}$·mg$^{-1}$ was measured with three independent hydrogenase-mediated benzyl viologen dichloride reduction assays.

### Preparation of Hyd-1/C catalyst
Catalyst preparation was carried out in a glove box (GLOVE BOX TECHNOLOGY LTD.) under a protective N$_2$ atmosphere (O$_2$ < 3 ppm). A 20 mg/mL carbon black suspension in PB (50 mM, pH 6.0 unless stated otherwise) was sonicated for 1 hour. For the preparation of the catalyst for one 2 mL scale reaction with 10 mM concentration of substrate, 52.8 μL of this suspension was transferred to an EPPENDORF tube, 15.4 μL of Hyd-1 solution (1.71 mg/mL) was added (C:Hyd-1 = 40:1 mass ratio), the mixture was gently mixed and left in the fridge (4 °C) for 1 hour. After that, the suspension of the catalyst was centrifuged (3 min, 14,100 × g), the supernatant was removed by pipetting, and the catalyst was resuspended in 100 μL of PB (50 mM, pH 6.0 unless stated otherwise). Resuspension-centrifugation-pipetting steps were repeated 3 times, and then the catalyst was resuspended in 100 μL of PB and then directly used for the reaction or frozen in liquid N$_2$ and stored at −80 °C.

### Small-scale hydrogenation reactions with Hyd-1/C catalyst
Reaction set-up was carried out in a glove box (GLOVE BOX TECHNOLOGY LTD.) under a protective N$_2$ atmosphere (O$_2$ < 3 ppm). Reactions were run on a 2 mL scale with a 10 mM concentration of substrate in PB (50 mM, pH 6.0 unless stated otherwise) or with 10% v/v of MeCN at room temperature under a gentle H$_2$ flow in an ASYNT OCTO MINI REACTOR, which allows running eight reactions in parallel (Supplementary Fig. 5). A stock solution of substrate in buffer or MeCN was transferred to a reaction vessel, 100 μL of catalyst was added, and the volume was adjusted with the corresponding buffer to a total volume of 2 mL. Substrates **8, 12, 14, 17–20, 23, 24, 26,** and **30** required double catalyst loading; for substrate **27** quadruple catalyst loading was used; substrate **28** required double catalyst loading and pH 8.0 (PB, 50 mM). Stock solutions of substrates **4, 8–13, 15–18, 20–24, 26–30** were prepared in MeCN so that the total amount of MeCN in the reaction mixture was 10% v/v. The reactor was closed and removed from the glove box. The hydrogen line was connected, and reactions were run at a 30-40 mL/min flow of H$_2$. Time points were taken at 24, 48, and

72 hours and analysed by $^1$H-NMR spectroscopy. To prepare a sample for the $^1$H-NMR analysis, an aliquot of 480 μL of the reaction mixture was centrifuged (3 min, 14,100 × $g$), 450 μL of supernatant was placed in the NMR tube, and 50 μL of D$_2$O was added.

## Gram-scale synthesis of procainamide using Hyd-1/C catalyst

Catalyst preparation was carried out in a glove box (GLOVE BOX TECHNOLOGY LTD.) under a protective N$_2$ atmosphere (O$_2$ < 3 ppm). A suspension of 264 mg of carbon black in 13.2 mL of PB (50 mM, pH 6.0) was sonicated in a 15 mL FALCON™ tube for 1 hour. After that, 1.02 mL of Hyd-1 solution (6.5 mg/mL) was added (C:Hyd-1 = 40:1 mass ratio), the mixture was gently mixed and left in the fridge (4 °C) for 1 hour. Next, the suspension of the catalyst was centrifuged (10 min, 11,309 × $g$), the supernatant was decanted, and the catalyst was resuspended in 25 mL of PB (50 mM, pH 6.0). Resuspension-centrifugation-decanting steps were repeated 5 times, and then the catalyst was resuspended in 25 mL of PB (50 mM, pH 6.0). Then, 50 mL of stock solution of substrate **31** in MeCN (100 mM) was transferred to a three-neck round-bottom flask (1 L) equipped with a stirring bar, the catalyst suspension was added, and the volume was adjusted to a total volume of 500 mL with the same buffer. The flask was closed with three SUBA-SEAL® septa and removed from the glove box. The hydrogen line was connected, and the reaction was run at a 30 mL/min flow of H$_2$ at 25 °C in a temperature-controlled oil bath. Reaction completion was confirmed after 24 hours by $^1$H-NMR spectroscopy. The reaction mixture was then transferred into a centrifuge bottle and centrifuged for 20 min at 11,309 × $g$. The supernatant was transferred into a separation funnel and washed with EtOAc (2 × 50 mL). The aqueous layer was collected into an Erlenmeyer flask and pH was adjusted to ~10-11 by slow addition of saturated aqueous K$_2$CO$_3$ solution followed by addition of brine (100 mL). The resulting solution was extracted with EtOAc (10 × 50 mL), combined organic layers were washed with brine (100 mL), dried with Na$_2$SO$_4$, filtered, and concentrated *in vacuo* to yield procainamide (**31a**) as a yellow oil.

## Recycling of Hyd-1/C catalyst

Reaction set-up was carried out in a glove box (GLOVE BOX TECHNOLOGY LTD.) under a protective N$_2$ atmosphere (O$_2$ < 3 ppm). Reactions were run on a 2 mL scale with a 10 mM concentration of **1** in PB (50 mM, pH 6.0) at room temperature under a gentle H$_2$ flow in an ASYNT OCTO MINI REACTOR. A stock solution of substrate in buffer was transferred to a reaction vessel, 100 μL of catalyst suspension in PB (50 mM, pH 6.0) was added, and the volume was adjusted to a total of 2 mL with the corresponding buffer. The reactor was closed and removed from the glove box. The hydrogen line was connected, and the reaction was run at a 30 mL/min flow of H$_2$. After 24 hours of reaction time, the reaction mixture was transferred to an EPPENDORF TUBE® and centrifuged (3 min, 14,100 × $g$), the supernatant was removed and analysed by $^1$H-NMR spectroscopy. The remaining catalyst was resuspended in 100 μL PB (50 mM, pH 6.0) and subjected to the next reaction cycle. The same procedure was repeated until a drop of conversion of the starting material was observed (Supplementary Fig. 43).

## Electrochemical procedures

All electrochemistry was conducted using a 3-electrode cell consisting of a pyrolytic graphite edge (PGE) working electrode (WE, 0.031 cm$^2$), a saturated calomel reference electrode (SCE, PALMSENS BV), a coiled platinum wire counter electrode (CE), and aqueous electrolyte (PB 50 mM, pH 6.0 unless otherwise specified). Cell temperature was kept constant at 25 °C by a water jacket. All experiments were conducted in a N$_2$ atmosphere (O$_2$ < 3 ppm) glovebox (GLOVE BOX TECHNOLOGY LTD.).

## Voltammetry of Hyd-1 and nitrobenzene

Cyclic voltammograms at stationary electrodes of nitrobenzene (**1**), *N*-phenylhydroxylamine (**1b**) and nitrosobenzene (**1c**) were recorded using a PalmSens4 potentiostat (PALMSENS BV) and the associated PSTrace 5.8 software. Cyclic voltammograms (CVs) were recorded with $n = 2$ replicates on independent electrodes and 3 sequential CVs recorded within each replicate. Each measurement included a 20 s poise to equilibrate the electrode before the scan. Aqueous electrolyte for all measurements was PB (50 mM, pH 6.0) containing 1 mM substrate (note: where solubility is <1 mM, this is a saturated solution). Each repeat included a control measurement in PB (50 mM, pH 6.0) without substrate to ensure a clean electrode surface before the start of each measurement. Before use, each WE was polished through 600, 1200 and 4000 grit sandpapers to achieve a high shine, with three 5-second pulses in a sonicator in MilliQ-water and rinsing with ethanol between each grade of sandpaper.

## Enzyme film voltammograms

To determine the onset potential of H$_2$ oxidation by Hyd-1 and Hyd-2 voltammograms were recorded using an AUTOLAB PGSTAT30 potentiostat (ECOCHEMIE) and the associated Nova 1.1 software. The cell headspace was purged with a constant flow of 1000 scc/min H$_2$ gas before and during measurements. Aqueous electrolyte for all measurements was PB (50 mM, pH 6.0). Voltammograms were recorded with $n = 2$ replicates on independent electrodes. Each experiment included a control measurement with no enzyme film. Before use, each WE was polished with 400 grit sandpaper and rinsed with MilliQ-water, and was then modified with an enzyme film by dropping 2 μL of enzyme solution (Hyd-1 = 1.71 mg/mL, Hyd-2 = 6.3 mg/mL) onto the electrode which was allowed to incubate for 5–10 min before rinsing again with MilliQ water. The modified electrode was mounted onto a rotator (METROHM) and rotated at 3000 rpm for the duration of the measurements to negate mass transport effects.

## Whole catalyst voltammograms

Whole catalyst voltammograms were recorded by the same method as the enzyme film voltammograms with the exception that 3 μL of freshly prepared catalyst suspension (Hyd-1/C) was drop-cast on to the polished electrode in place of the enzyme.

## Determination of substrate reduction onset potentials

A broader study of the reduction onset potentials in aqueous electrolyte of nitro-containing compounds was conducted to assess whether these thermodynamic predictions of catalyst reactivity were accurate. Since these are not currently available in literature under aqueous conditions, we used voltammetry of the substrates under reaction conditions to determine these.

Substrate reduction voltammograms were recorded using a PalmSens4 (PALMSENS BV) potentiostat and the associated PSTrace 5.8 software. Cyclic voltammograms (CVs) were recorded with $n = 2$ replicates on independent electrodes and 3 sequential CVs recorded within each replicate following a 20 s poise to equilibrate the electrode. The first sweep of each (equivalent to a linear sweep voltammogram) was taken for analysis. Electrolyte for all measurements was PB (50 mM, pH 6.0) containing 1 mM substrate (note: where solubility is <1 mM, this is a saturated solution). Each repeat included a control measurement in PB (50 mM, pH 6.0) without substrate. Before use, each WE was polished through 600, 1200 and 4000 grit sandpapers to achieve high shine, with three 5 s pulses in MilliQ-water in a sonicator and rinsing with ethanol between each grade of sandpaper. Working electrodes were then mounted onto an AutoLab (METROHM) rotator and were rotated at 3000 rpm, unless otherwise specified, during data acquisition to negate mass transport effects, therefore providing a study of substrate behaviour at the carbon surface.

Onset potentials were then calculated by analysis of the first voltammogram for each substrate once the electrode was placed into solution. Checking for consistency with the control voltammogram taken before each measurement, a linear baseline correction was

extrapolated from a region of no activity. The potential at which the difference between the measured current and the baseline began to increase exponentially, exceeding a threshold value of 10 nA (instrument resolution -0.5 nA), was taken to be the onset potential. Each voltammogram and calculation was repeated with two independent electrodes to ensure true repeats on newly prepared carbon surface.

## Small-scale hydrogenation reactions with Hyd-2/C catalyst

Catalyst preparation was carried out in a glove box (GLOVE BOX TECHNOLOGY LTD.) under a protective $N_2$ atmosphere ($O_2 < 3$ ppm). A 20 mg/mL carbon black (BP2000, CABOT) suspension in PB (50 mM, pH 6.0) was sonicated for 1 hour. For the preparation of the catalyst loading for one 1 mL scale reaction with 10 mM concentration of substrate, 7.5 μL of this suspension was transferred to an EPPENDORF tube, 2.13 μL of Hyd-2 solution (6.3 mg/mL) was added (C:Hyd-2 = 5.6:1 mass ratio), the mixture was gently mixed and left on ice for 1 hour. After that, the suspension of the catalyst was centrifuged (3 min, 14,100 × g), the supernatant was removed, and the catalyst was resuspended in 50 μL of PB (50 mM, pH 6.0). The catalyst was then directly added to the reaction vial. Reactions were run on a 0.5 mL scale with 10 mM of substrate in PB (50 mM, pH 6.0) in a pressure vessel reactor, charged to 2 bar $H_2$ and placed on a rocker to facilitate mixing. For $^1$H-NMR analysis, the reaction mixture was centrifuged to collect the catalyst particles, 450 μL of supernatant and 50 μL of $D_2O$ were added to the NMR tube. $^1$H-NMR spectra were recorded with water signal suppression method.

## Data availability

All data supporting the findings of this study are provided within the paper and its Supporting Information files. Correspondence and requests for materials should be addressed to K.A.V.

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

## Acknowledgements

We are grateful to Dr Stephen Carr, Alison Tam, and Plastida Synteng Umwi for assistance with isolation and purification of *E. coli* hydrogenases, and to Maya Landis for helpful discussion. We are grateful to Matthew Harris who carried out preliminary experiments to test hydrogenation of nitro compounds with J.S.R., H.A.R. and K.A.V. during a summer studentship supported by the Biotechnology and Biological Sciences Research Council, BBSRC. K.A.V., T.S., J.S.R and G.S. were supported by ERC CoG-819580 (BiocatSusChem, to K.A.V), and K.A.V. is additionally supported by BBSRC grant BB/X002624/1. The UK Catalysis Hub is kindly thanked for resources and support provided via membership of the UK Catalysis Hub Consortium (T.S.) and financial support for the work of T.S. by a project grant under EP/M013219/1 (biocatalysis) (to K.A.V). T.C.L. is grateful for the support of an EPSRC studentship. J.S.R. is grateful for the support of a Linacre College EPA Cephalosporin Junior Research Fellowship. Work of D.S. and K.A.V. was supported financially by the Innovation Centre for Sustainable Technologies (iCAST). K.A.V., J.S.R., H.A.R., and S.E.C. carried out some of the research towards this study with support from EPSRC IB Catalyst grant EP/N013514/1 to K.A.V. and H.A.R.

## Author contributions

Conceptualisation of the research was contributed by K.A.V., J.S.R., H.A.R., S.E.C, T.S., and D.S. Supervision of research was performed by K.A.V, H.A.R., S.E.C., J.S.R., T.S. and D.S. Experimental investigations were conducted by J.S.R, S.E.C., G.S., T.S., D.S. and T.C.L. Writing the original draft was contributed by K.A.V. and D.S. Revisions to the manuscript were carried out by all authors. Funding acquisition was contributed by K.A.V. and H.A.R.

## Competing interests

The authors declare the following competing interests: a patent application has been filed related to the catalyst system described in this manuscript (WO2023218206) and might afford royalties to the authors. D.S., J.S.R., G.S., and T.S. declare no further competing interests. Intellectual property covered by WO2023218206 is licensed to HydRegen. K.A.V, H.A.R. and S.E.C. are founders of HydRegen. T.C.L., H.A.R., and S.E.C. are current employees of HydRegen.
