## [Peer Review File · Nature Communications]

Selective hydrogenation of nitro compounds to amines by coupled redox reactions over a heterogeneous biocatalystEditorial note: Parts of this Peer Review File have been redacted as indicated to remove third-party material where no permission to publish could be obtained.

REVIEWER COMMENTS

Reviewer #1 (Remarks to the Author):

The manuscript presented by Sokolova et al further expands the application of the Vincent group hydrogenases, with the direct reduction of nitroarenes. The manuscript demonstrates that when immobilised on carbon black particles, as per the standard procedure from the Vincent group (refs 18-21 in this manuscript). The difference here is the hydrogenase is not coupled to another enzyme and recycling nicotinamide, it is performing direct catalysis itself through full reduction of the nitroarenes to the respective anilines. This impressive chemistry which is not directed through a co-factor (flavin/nicotinamide) seems impressive to me. It seems to highlight this enzyme as being extremely versatile as the reduction seems to be decoupled from the enzyme active site and happening on the surface of the enzyme? In the section discussing mechanism perhaps the authors could speculate as the actual location of the reduction, as this could potentially allow for any substrate to be reduced, so long as the potentials are correct? Either way, the impressive nature of the reported work warrants publication here. I have a few queries regarding the reporting of the work and some of the claims made, but other than that think this should be accepted pending minor revision.

- In the introduction there should be mention made of Poelerands recent work which has shown that a wild type NR can also effect full reduction to aniline from nitro (10.1002/cbic.202300846, 10.1038/s41467-023-41194-w). The earlier work also reports aliphatic nitro reduction too.

- I don't understand the need for the repetition of the two schemes of substrates, why can't the information simply be included in one of them? People should be able to work out what the starting materials and products both look like.

- More information should be included about how the reaction is set up. TTNs are reported but nothing is included around enzyme loading, despite there seeming to be some information in the SI. Is the reaction loading 1.71 mg mL⁻¹ or 1.32 mg mL⁻¹? It is written in quite an elusive way in the SI. I am not disputing any results, it would just be nice for this to be clearer for the reader.

-Leading on from that point, are the yields that are calculate averages? On the scales quoted in the SI, it would be hard to accurately verify those consistently IMO. Going to µg point is so hard, I think this should be duplicates at a minimum, if not triplicate. If you actually look at a lot of the NMRs there is still DMSO peaks in them, which infers to me that some of the yields could be inaccurate. Alternatively, could 3-5 of the substrates be scaled up to 1 mmol scale and yields declared for those examples instead?

- Further from that, some of the ¹H NMRs in the SI don't look of high enough standard (see DMSO comment above). S13 looks really strange around 2ppm for example. In addition S15 (NO₂), S19 (NH₂), Both S20, S21 could be obtained in DMSO/CDCl₃?, S24 (NH₂), S27 (NO₂, not pure?), S29, S32 (NO₂), S33,

Reviewer #2 (Remarks to the Author):

The manuscript entitled "Selective hydrogenation of nitro compounds to amines by coupled redox reactions over a heterogeneous biocatalyst" focuses on developing a new method for the heterogeneous biocatalyzed reduction of nitro compounds, a transformation typically accomplished either by expensive metal catalysis, often suffering from selectivity issues, or using nitroreductases that require the use of expensive cofactor(s). The authors tested their methodology on a wide panel of nitroarenes and full conversion and good yield observed. They also proved that the system avoids unwanted side reactions such as dehalogenation and reduction of unsaturated bonds. NMR and cyclic voltammetry studies provided mechanistic insights into the transformation. Moreover, proof of recyclability was provided, and the system proved to be stable for the first 5 cycles. The authors also presented preliminary proof of concept for aliphatic reduction using a different Hydrogenase enzyme, which is more suitable for these substrates due to the lower onset potential required.

The manuscript presents an interesting contribution to the field of coupled redox reactions with an heterogeneous biocatalyst. The novelty lies mainly in the method presented: the integration of hydrogenase enzymes on solid supports to enable cofactor-free catalysis demonstrating high chemo-selectivity, stability, and reusability and turnovers. However, it is important to note that the products can also be readily synthesized using oxidoreductases and/or transaminase enzymes under mild conditions. Additionally, the developed method does not present any potential to be applied for stereoselective transformations.

My main concerns are:

- The references 1-3 do not comprehensively cover the methods for efficient and sustainable synthesis of amines. Over the past decade, biocatalytic amine synthesis has been extensively studied, resulting in the development of enzymes capable of synthesizing primary, secondary, and tertiary amines with complete enantioselectivity. These enzymes include AmDHs, RedAms, IReds, and ω -TAs. While a few examples are provided below, I recommend that the authors include additional references to better illustrate this point.

Nat Catal 2, 324–333 (2019). <https://doi.org/10.1038/s41929-019-0249-z>

Nat Commun 15, 4933 (2024). <https://doi.org/10.1038/s41467-024-49009-2>

Nat Commun 10, 3717 (2019). <https://doi.org/10.1038/s41467-019-11509-x>

Nature 604, 86–91 (2022). <https://doi.org/10.1038/s41586-022-04458-x>

ACS Catal. 2018, 8, 10985–11015. [10.1021/acscatal.8b02924](https://doi.org/10.1021/acscatal.8b02924)

- Figure 2: One of the two experiments was performed under CO₂ at pH 6. What was the pH for the other experiment? If a different pH was used, the potential should be referenced against RHE rather than SHE. Additionally, please explain how you calculated the onset potential. It appears that the lines are not drawn at the beginning of the catalytic slope.

- The authors state they selected their nickel-iron (NiFe) hydrogenase enzyme based on its stability and oxygen tolerance (page 3, line 65). However, the catalyst synthesis and the reaction were prepared under nitrogen atmosphere (inside a glove box). Can they comment on that or provide experimental proof of the feasibility of the reaction under atmospheric conditions?

- The authors report the synthesis of a pharmaceutically relevant molecule and stated (page

5 line 140) that “This clearly indicates scalability and potential applicability of this system in the production of fine chemicals and their precursors.” Perhaps it would be appropriate to add a reference to the current established method for synthesizing this compound to prove or reinforce this statement, also considering that the purity is lower than the requirements for pharmaceuticals. Alternatively, they may rephrase the sentence.

- In the conclusion section, the authors claim, “inspired by the concept of electrochemical hydrogenation in heterogeneous catalysis, we have established a hybrid bio-chemo catalyst, which operates entirely via an electrochemical (coupled redox) mechanism” (page 8, line 184). This statement might be misleading, because the reaction is not fundamentally electrochemical.
- In the supporting information (page S50) the authors reported recycling experiments which reinforce the mechanistic hypothesis and the robustness of the system. However, they do not comment extensively on the deactivation mechanism. Is the loss of activity and the increase of hydroxylamine intermediate due to the enzyme deactivation (which may release fewer electrons therefore making the transformation slower) or because of the partial poisoning of the carbon black support? Moreover, can they comment on the apparent increase of the product amount after the 13th cycle?
- Given the limitations of the substrate scope to nitroarenes, why did the authors decided to focus on the Hyd-1 enzyme and did not develop a system based on the Hyd-2 enzymes that possibly display a broader scope?
- Lines 106-109: The authors may provide an explanation for the observed selectivity that will help the general readership.

Reviewer #3 (Remarks to the Author):

Response to Referees' Comments

for 'Selective hydrogenation of nitro compounds to amines by coupled redox reactions over a heterogeneous biocatalyst'

by Sokolova et al.

We were very pleased to see that the reviewers considered that our manuscript reports 'impressive chemistry' which 'presents an interesting contribution to the field of coupled redox reactions', and we thank them for their comments. We have addressed all the comments of the reviewers in detail as described below, making modifications to the main text or the SI as required. We hope that with these changes, the manuscript will be suitable for publication in Nature Communications.

Reviewer #1 (Remarks to the Author):

The manuscript presented by Sokolova et al further expands the application of the Vincent group hydrogenases, with the direct reduction of nitroarenes. The manuscript demonstrates that when immobilised on carbon black particles, as per the standard procedure from the Vincent group (refs 18-21 in this manuscript). The difference here is the hydrogenase is not coupled to another enzyme and recycling nicotinamide, it is performing direct catalysis itself through full reduction of the nitroarenes to the respective anilines. This impressive chemistry which is not directed through a co-factor (flavin/nicotinamide) seems impressive to me. It seems to highlight this enzyme as being extremely versatile as the reduction seems to be decoupled from the enzyme active site and happening on the surface of the enzyme? In the section discussing mechanism perhaps the authors could speculate as the actual location of the reduction, as this could potentially allow for any substrate to be reduced, so long as the potentials are correct? Either way, the impressive nature of the reported work warrants publication here. I have a few queries regarding the reporting of the work and some of the claims made, but other than that think this should be accepted pending minor revision.

→ The nitro reduction half reaction is occurring on the surface of the carbon support. Control experiments with hydrogenase alone in solution showed no conversion of the substrate nitro benzene (Section III of the SI, Fig. S3), indicating the critical role of the carbon as a functional support, probably in accumulating the six electrons needed for nitro reduction. As such, the catalyst concept could possibly be extended to other substrates with appropriate potentials, but carbon is known for being inert to many catalytic processes (hence its frequent application as an innocent electrode material in studies of other immobilised catalysts), and the nitro hydrogenation is unusual in occurring readily at the carbon surface. This is actually a strong advantage for our catalyst (compared to broad specificity hydrogenation catalysts like Pd or Pt) because it results in a high catalytic selectivity for the nitro functional group over other unsaturated bonds (carbonyl, alkene, alkyne, nitrile) which we show are not converted by this catalyst.

In response to a related comment from Reviewer 2 (see below), we have added an additional sentence relating to selectivity in the manuscript after the paragraph lines 106-109,

'The selectivity is consistent with the clean linear sweep voltammograms observed for electrochemical reduction of these nitroarenes at carbon.'

- In the introduction there should be mention made of Poelerands recent work which has shown that a wild type NR can also effect full reduction to aniline from nitro (10.1002/cbic.202300846, 10.1038/s41467-023-41194-w). The earlier work also reports aliphatic nitro reduction too.

→ We recognise that there are some examples of nitro-reductase enzymes converting the nitro group right through to the amine, and had acknowledged this with the wording, 'often fail to progress beyond the N-hydroxylamine intermediate...'. We have now updated this to

'and with few exceptions,¹⁵ often fail to progress beyond the N-hydroxylamine intermediate',

where reference 15 is the very recent manuscript from Poelerands and coworkers, 10.1002/cbic.202300846.

We had also referred to several methods which have been shown to improve conversion to the amine, including photocatalysis, and hence we had already referenced the second manuscript (10.1002/cbic.202300846) from Poelerands and coworkers in that context as reference 10 in the former numbering, now reference 16 in the updated manuscript,

'although this can be mitigated by photocatalysis¹⁶'

- I don't understand the need for the repetition of the two schemes of substrates, why can't the information simply be included in one of them? People should be able to work out what the starting materials and products both look like.

→ Figures 3 and 4A of the manuscript are provided to present different sets of data, rather than just to show reactant and product. Figure 3 indicates the time taken to achieve full conversion for each nitro substrate from initial small-scale screening reactions, whereas Figure 4A shows the isolated yield obtained for each amine product from a separate set of reactions. We feel strongly that it is useful to retain both of these figures, and it would confuse the reader to combine them as we refer to both substrates and products in the text.

- More information should be included about how the reaction is set up. TTNs are reported but nothing is included around enzyme loading, despite there seeming to be some information in the SI. Is the reaction loading 1.71 mg mL⁻¹ or 1.32 mg mL⁻¹? It is written in quite an elusive way in the SI. I am not disputing any results, it would just be nice for this to be clearer for the reader.

→ Section II of the SI already describes the catalyst preparation procedure in detail:

'Catalyst preparation was carried out in a glove box (GLOVE BOX TECHNOLOGY LTD.) under a protective N₂ atmosphere (O₂ < 3 ppm). A 20 mg/mL carbon black suspension in PB (50 mM, pH 6.0 unless stated otherwise) was sonicated for 1 hour. For the preparation of the catalyst for one 2 mL scale reaction with 10 mM concentration of substrate, 52.8 μL of this suspension was transferred to an Eppendorf tube, 15.4 μL of Hyd-1 solution (1.71 mg/mL) was added (C:Hyd-1 = 40:1 mass ratio), the mixture was gently mixed and left in the fridge (4 °C) for 1 hour. After that, the suspension of the catalyst was centrifuged (3 min, 14500 rpm), the supernatant was removed by pipetting, and the catalyst was resuspended in 100 μL of PB (50 mM, pH 6.0 unless stated otherwise). Resuspension-centrifugation-pipetting steps were repeated 3 times, and then the catalyst was resuspended in 100 μL of PB and then directly used for the reaction or frozen in liquid N₂ and stored at -80 °C.

Note: after numerous preliminary screenings the optimal ratio of carbon to Hyd-1 (C:Hyd-1) to reach full conversion of various substrates to corresponding amines was found to be **40:1 mass ratio** with corresponding catalyst loading of **1.32 mg of Hyd-1 per 1 mmol of substrate.**'

For further clarity, we have also added into the captions for Figure 3 and Figure 4 of the manuscript the mass of Hyd1 and carbon used for running each reaction:

'**Figure 3.** Substrate scope of hydrogenation reactions achieved with the Hyd-1/C catalytic system (unless specified, single catalyst loading: 1.06 mg of C, 26 µg of Hyd-1 per reaction) at 10 mM concentration of substrate, 2 mL reaction volume, 0% or 10% v/v % of MeCN in sodium phosphate buffer (PB, 50 mM, pH 6.0, unless stated otherwise), room temperature, 1 bar H₂. *Double catalyst loading. **Quadruple catalyst loading. #Double catalyst loading, pH 8.0.'

'**Figure 4. A.** Isolated yields (%) for **1a–30a**. Conditions: as in Figure 3 caption. *¹H-NMR yields. **B.** ¹H-NMR traces of hydrogenation of **1** at indicated time points. Conditions: Hyd-1/C (1.06 mg of C, 26 µg of Hyd-1), 10 mM **1**, 2 mL reaction volume, PB (50 mM, pH 6.0), room temperature, 1 bar H₂. Traces for **1**, **1b**, and **1a** are labelled with red, green, and blue, respectively. **C.** Synthesis of substrate **31** and its hydrogenation on a gram-scale. Conditions: Hyd1/C (264 mg of C, 6.63 mg of Hyd-1), 10 mM **31**, 500 mL reaction volume, 10% MeCN v/v% in PB (50 mM, pH 6.0), room temperature, 1 bar H₂.'

-Leading on from that point, are the yields that are calculate averages? On the scales quoted in the SI, it would be hard to accurately verify those consistently IMO. Going to µg point is so hard, I think this should be duplicates at a minimum, if not triplicate. If you actually look at a lot of the NMRs there is still DMSO peaks in them, which infers to me that some of the yields could be inaccurate. Alternatively, could 3-5 of the substrates be scaled up to 1 mmol scale and yields declared for those examples instead?

→ Isolation experiments were carried out on a mg scale, not µg as the reviewer suggested, with isolated yields reported (actual mass of the product weighed after isolation) except for 2 products (**18a**, **25a**), for which the NMR yields are reported due to the instability of these products which resulted in difficulties with their isolation. (This point is noted clearly in the main text, and these 2 products are indicated by * in Figure 4A.) The characteristic ¹H NMR signals of all the isolated products are compared with literature data and are summarised in Section V of the SI.

The ¹H NMR spectra reported in Section IV of the SI represent spectra of the starting materials compared to the reaction mixture spectra after the completion of each reaction, with results summarised in Figure 3 of the main text, therefore some spectra contain signals of co-solvent, acetonitrile (not DMSO), which was required to dissolve some substrates. As described in Section IV of the SI,

'Time points were taken at 24, 48, and 72 hours and analysed by ¹H-NMR spectroscopy. To prepare a sample for the ¹H-NMR analysis, an aliquot of 480 µL of the reaction mixture was centrifuged (3 min, 14500 rpm), 450 µL of supernatant was placed in the NMR tube, and 50 µL of D₂O was added. ¹H-NMR spectra (Fig. S6-S35) were measured with water signal suppression. The signal observed around 2 ppm for some of the ¹H-NMR spectra corresponds to MeCN.'

To make it clearer we added a sentence to line 145 of the SI,

¹H-NMR spectra (Fig. S6-S35) were measured with water signal suppression and represent comparison of signals of each starting material with the corresponding amine product after completion of each reaction.'

The experiments shown in Figure 3 were used to define conditions for the isolated yield measurements (summarised in Figure 4A), for which isolated product ¹H NMR spectra were recorded and verified by comparison to literature data. This is all summarised in Section V of the SI.

- Further from that, some of the 1H NMRs in the SI don't look of high enough standard (see DMSO comment above). S13 looks really strange around 2ppm for example. In addition S15 (NO₂), S19 (NH₂), Both S20, S21 could be obtained in DMSO/CDCl₃?, S24 (NH₂), S27 (NO₂, not pure?), S29, S32 (NO₂), S33,

→ An explanation has been provided in the response to the comment above. Characteristic signals of products are reported in CDCl₃, DMSO-d₆ or methanol-d₄ (Section V of the SI).

Reviewer #2 (Remarks to the Author):

The manuscript entitled "Selective hydrogenation of nitro compounds to amines by coupled redox reactions over a heterogeneous biocatalyst" focuses on developing a new method for the heterogeneous biocatalyzed reduction of nitro compounds, a transformation typically accomplished either by expensive metal catalysis, often suffering from selectivity issues, or using nitroreductases that require the use of expensive cofactor(s). The authors tested their methodology on a wide panel of nitroarenes and full conversion and good yield observed. They also proved that the system avoids unwanted side reactions such as dehalogenation and reduction of unsaturated bonds. NMR and cyclic voltammetry studies provided mechanistic insights into the transformation. Moreover, proof of recyclability was provided, and the system proved to be stable for the first 5 cycles. The authors also presented preliminary proof of concept for aliphatic reduction using a different Hydrogenase enzyme, which is more suitable for these substrates due to the lower onset potential required. The manuscript presents an interesting contribution to the field of coupled redox reactions with an heterogeneous biocatalyst. The novelty lies mainly in the method presented: the integration of hydrogenase enzymes on solid supports to enable cofactor-free catalysis demonstrating high chemo-selectivity, stability, and reusability and turnovers. However, it is important to note that the products can also be readily synthesized using oxidoreductases and/or transaminase enzymes under mild conditions. Additionally, the developed method does not present any potential to be applied for stereoselective transformations.

→ The majority of substrates demonstrated here are nitroarenes which obviously offer no scope for stereoselective transformations. Further study of nitroaliphatic substrates is underway, but was beyond the scope of this study because of the limited availability of Hyd-2 which is more difficult to prepare than Hyd-1, as discussed below.

My main concerns are:

- The references 1-3 do not comprehensively cover the methods for efficient and sustainable synthesis of amines. Over the past decade, biocatalytic amine synthesis has been extensively studied, resulting in the development of enzymes capable of synthesizing primary, secondary, and tertiary amines with complete enantioselectivity. These enzymes include AmDHs, RedAms, IReds, and ω -TAs. While a few examples are provided below, I recommend that the authors include additional references to better illustrate this point.

Nat Catal 2, 324–333 (2019). <https://doi.org/10.1038/s41929-019-0249-z>

Nat Commun 15, 4933 (2024). <https://doi.org/10.1038/s41467-024-49009-2>

Nat Commun 10, 3717 (2019). <https://doi.org/10.1038/s41467-019-11509-x>

Nature 604, 86–91 (2022). <https://doi.org/10.1038/s41586-022-04458-x>

ACS Catal. 2018, 8, 10985–11015. 10.1021/acscatal.8b02924

→ Our introduction fairly quickly focusses down on nitro reductions as a route to amines because of the vast literature on amine synthesis. Therefore, we had referenced biocatalytic approaches to nitro reduction but had not explicitly covered other biocatalytic routes to amines. However, to address the reviewer's comment, we have expanded the second sentence of the main text to read,

'This has led to a wide range of developments in selective methods for amine synthesis,¹⁻³ including various biocatalytic approaches.⁴⁻⁸'

We have therefore added in the suggested references as 4-8.

- Figure 2: One of the two experiments was performed under CO₂ at pH 6. What was the pH for the other experiment? If a different pH was used, the potential should be referenced against RHE rather than SHE. Additionally, please explain how you calculated the onset potential. It appears that the lines are not drawn at the beginning of the catalytic slope.

→ The voltammogram in Figure 2A was recorded under a N₂ atmosphere, while the voltammogram in panel B was recorded under a H₂ atmosphere. Both voltammograms were recorded at pH 6.0. We have updated the caption to Figure 2 to correct and clarify this,

Figure 2. Onset potential for nitrobenzene reduction and H₂ oxidation on a carbon electrode at 25 °C, pH 6.0. Cyclic voltammograms for **A**: nitrobenzene at a stationary graphite electrode under a N₂ atmosphere, scan rate 10 mV/s; and **B**: a film of Hyd-1 adsorbed onto the electrode under a H₂ atmosphere with electrode rotation at 3000 rpm, scan rate 1 mV/s. Potentials are quoted vs the standard hydrogen electrode, SHE. Dashed vertical line: potential of the 2H⁺/H₂ couple at the experimental conditions, $E'(2H^+/H_2)$; solid black vertical line: measured onset potential for H₂ oxidation by Hyd-1; solid grey vertical line: measured onset for nitrobenzene reduction (see SI Table S6).'

It is common practice in bioelectrochemistry to reference the potential to SHE (i.e. $E_{SHE} = 0$ V at pH 0), even when reactions are carried out in water around neutral pH values. We state the potential of the H⁺/H₂ couple at pH 6.0 in the manuscript,

'At pH 6.0, 1 bar H₂, the potential of the proton/dihydrogen couple, $E'(2H^+/H_2)$ is -0.355 V.'

In Figure 2 both voltammograms were recorded at pH 6.0, and we plot the two voltammograms directly above each other on the same scale (ie vs SHE) to make it easy for the reader to compare.

All subsequent reactions and electrochemistry were carried out at pH 6, with the exception of reactions of substrate **28**, where we explicitly note in the caption of Figure 3: '#Double catalyst loading, pH 8.0'. We have added clarification of this in the caption to Figure 4: 'except substrate **28**: pH 8.0'.

Regarding onset potentials: detailed description of the electrochemical method and the data processing for calculation of onset potentials for the nitro-group reductions are provided in the SI, section IX.3. The description of data processing has been very slightly modified for clarity, as follows:

'Onset potentials were then calculated by analysis of the first voltammogram for each substrate once the electrode was placed into solution. Checking for consistency with the control voltammogram taken before each measurement, a linear baseline correction was extrapolated from a region of no activity. The potential at which the difference between the measured current and the baseline began to increase exponentially, exceeding a threshold value of 10 nA (instrument resolution ~0.5 nA), was taken to be the onset potential. Each voltammogram and calculation was repeated with two independent electrodes to ensure true repeats on newly prepared carbon surface and both data points are reported in Figure S48.'

We thank the reviewer for the query regarding the onset potentials marked on Figure 2. The onset potential for nitrobenzene reduction marked on Figure 2 is the value quoted in Table S6 of the SI. All of the onset potential values in Table S6 were determined from linear sweep voltammograms (eg shown in Figures S49-S74 of the SI), and were recorded with the electrode rotating, because a more defined and sharper onset was observed under these conditions. A sluggish commencement of the nitro reduction wave means that the onset potential is not obvious to the eye without zooming in to the voltammetric data in Figure 2. We have clarified how this onset potential was calculated by amending the caption to Figure 2:

'measured onset for nitrobenzene reduction (see SI Table S6).'

- The authors state they selected their nickel-iron (NiFe) hydrogenase enzyme based on its stability and oxygen tolerance (page 3, line 65). However, the catalyst synthesis and the reaction were prepared under nitrogen atmosphere (inside a glove box). Can they comment on that or provide experimental proof of the feasibility of the reaction under atmospheric conditions?

→ A clear benefit of the O₂ tolerance of Hyd-1 for biotechnology is that it makes isolation of the enzyme significantly easier since it can be isolated and handled aerobically. A H₂ atmosphere is required in any case for the hydrogenation reactions, so reactions would always be set up under H₂. During preliminary reaction screening we observed that it is possible to set up the reactions on the bench under a H₂ flow, although we noticed that reactions were slower for some substrates in this case.

- The authors report the synthesis of a pharmaceutically relevant molecule and stated (page 5 line 140) that “This clearly indicates scalability and potential applicability of this system in the production of fine chemicals and their precursors.” Perhaps it would be appropriate to add a reference to the current established method for synthesizing this compound to prove or reinforce this statement, also considering that the purity is lower than the requirements for pharmaceuticals. Alternatively, they may rephrase the sentence.

→ The reviewer is correct in noting that the purity of the procainamide (96%), which was synthesised by hydrogenation of its precursor using our catalytic system, is lower than the level required for pharmaceuticals. However, we would like to particularly highlight that in this study procainamide was obtained with 96% purity by simple isolation of the product from the reaction mixture without any further purification. In an industrial pharmaceutical setting it would be typical to apply standard purification procedures, often using preparative HPLC, to reach the required purity. We deliberately reported the purity of the as-isolated compound to demonstrate the high selectivity of our catalytic approach.

- In the conclusion section, the authors claim, “inspired by the concept of electrochemical hydrogenation in heterogeneous catalysis, we have established a hybrid bio-chemo catalyst, which operates entirely via an electrochemical (coupled redox) mechanism” (page 8, line 184). This statement might be misleading, because the reaction is not fundamentally electrochemical.

→ The use of the term ‘electrochemical’ to refer to a process which proceeds via coupled redox half reactions, as in our catalyst, is becoming well-established in the field of heterogeneous catalysis, and we use these terms here deliberately to make a link to this field. Recognising the possibility for confusion with readers less familiar with the heterogeneous catalysis literature, we had already added ‘coupled redox’ in parenthesis to make sure this was clear. As an example of the prominence of this terminology in heterogeneous catalysis literature, we show, below, an extract from a recent publication from the group of Surendranath where this terminology is used several times (Nature Catalysis, <https://doi.org/10.1038/s41929-023-01094-0>, cited as reference number 20 in the new numbering in our manuscript). For a broad interest journal such as Nature Communications, we feel that it is important to use terminology that makes clear the links and commonalities between different fields of research and show how each field may be moved forward into new territory by learning from tangential fields:

[text redacted]

- In the supporting information (page S50) the authors reported recycling experiments which reinforce the mechanistic hypothesis and the robustness of the system. However, they do not comment extensively on the deactivation mechanism. Is the loss of activity and the increase of hydroxylamine intermediate due to the enzyme deactivation (which may release fewer electrons therefore making the transformation slower) or because of the partial poisoning of the carbon black support? Moreover, can they comment on the apparent increase of the product amount after the 13th cycle?

→ The loss of activity of the catalyst in recycling experiment can most likely be attributed to the small scale of this experiment and the problem of losing some catalyst after each cycle when separating the catalyst by centrifuging reaction mixture and decanting the supernatant after centrifugation. We believe that larger scale reactions will lower the impact of loss of the catalyst and will improve catalyst

performance during recycling. Subsequent experiments have shown that lowering the catalyst loading results in appearance of hydroxylamine in the product, and hence catalyst loss is the most likely explanation for the product distribution in later reuse cycles. The apparent increase of the product amount after cycle 13 can be attributed as an artefact as it was a small-scale experiment and there could be some fluctuations and/or integration errors. We have added a sentence into the main text to address the catalyst loss question:

‘The small scale of these reactions (2 mL reaction volume) means that catalyst loss during each recovery cycle is more significant than it would be in larger scale reactions.’

- Given the limitations of the substrate scope to nitroarenes, why did the authors decided to focus on the Hyd-1 enzyme and did not develop a system based on the Hyd-2 enzymes that possibly display a broader scope?

→ Hyd-1 is much easier to isolate than Hyd-2 because we have a modest overexpression system for Hyd-1. It is also more stable than Hyd-2, showing activity over many days. The results with Hyd-2 are therefore conceptual, but difficult to scale because of the limited availability of Hyd-2.

- Lines 106-109: The authors may provide an explanation for the observed selectivity that will help the general readership.

→ We discuss this in response to the first comment from reviewer 1. Briefly, carbon is known for being inert to many catalytic processes (hence its frequent application as an innocent electrode material in studies of other immobilised catalysts), and the nitro hydrogenation is unusual in occurring readily at the carbon surface. This is actually a strong advantage for our catalyst (compared to broad specificity hydrogenation catalysts like Pd or Pt) because it results in a high catalytic selectivity for the nitro functional group over other unsaturated bonds (carbonyl, alkene, alkyne, nitrile) which we show are not converted by our catalyst. To clarify this point in the manuscript, as noted above, we have added an additional sentence to the paragraph approx. lines 106-109,

‘The selectivity is consistent with the clean linear sweep voltammograms observed for electrochemical reduction of these nitroarenes at carbon.’

Reviewer #3 (Remarks to the Author):

→ We thank reviewer 3 for co-reviewing the manuscript.

REVIEWERS' COMMENTS

Reviewer #1 (Remarks to the Author):

Comments addressed on my part for both reviewers. Well done on a great piece of work.

Reviewer #2 (Remarks to the Author):

I would like to thank the authors for their revision. I am satisfied with the changes made and the explanations provided. The authors have satisfactorily addressed the main technical concerns raised. They provided explanations regarding the voltammograms and the observed selectivity, which helped clarify the experimental conditions and outcomes. They also offered a detailed explanation for the catalyst's loss of activity in recycling experiments, attributing it to the small scale of the experiments.

Additionally, the authors acknowledged other existing methods for amine synthesis and cited relevant examples, demonstrating an awareness of the current state of the field. They also recognized a recently reported effective method for biocatalytic nitro reduction, showing a balanced understanding of their work in the broader context. The authors explained the issue of the low purity of the obtained procainamide by emphasizing that this result was achieved after a simple work-up process. Furthermore, they clarified their choice of the Hyd-1 enzyme and provided additional experimental details, including: NMR spectra, catalyst loading, and preparation methods. These additions enhance the clarity and reproducibility of their work.

Reviewer #3 (Remarks to the Author):

I co-reviewed this manuscript with one of the reviewers who provided the listed reports.

This is part of the Nature Communications initiative to facilitate training in peer review and to provide appropriate recognition for Early Career Researchers who co-review manuscripts.